# Effectiveness of Booster Dose of Anti SARS-CoV-2 BNT162b2 in Cirrhosis: Longitudinal Evaluation of Humoral and Cellular Response

**DOI:** 10.3390/vaccines10081281

**Published:** 2022-08-08

**Authors:** Vincenzo Giambra, Annarita Valeria Piazzolla, Giovanna Cocomazzi, Maria Maddalena Squillante, Elisabetta De Santis, Beatrice Totti, Chiara Cavorsi, Francesco Giuliani, Nicola Serra, Alessandra Mangia

**Affiliations:** 1Institute for Stem Cell Biology, Regenerative Medicine and Innovative Therapies, Fondazione IRCCS “Casa Sollievo della Sofferenza”, 71013 San Giovanni Rotondo, Italy; 2Liver Unit, Fondazione IRCCS Ospedale “Casa Sollievo della Sofferenza”, 71013 San Giovanni Rotondo, Italy; 3ICT Innovation and Research Unit, Fondazione IRCCS Ospedale “Casa Sollievo della Sofferenza”, 71013 San Giovanni Rotondo, Italy; 4Department of Public Health, University “Federico II”, 80138 Napoli, Italy

**Keywords:** liver cirrhosis, SARS-CoV-2, mRNA vaccine, MELD score

## Abstract

**Background:** LC has been associated with hyporesponsiveness to several vaccines. Nonetheless, no data on complete serological and B- and T-cell immune response are currently available. **Aims:** To assess, in comparison with healthy controls of the same age and gender, both humoral and cellular immunoresponses of patients with LC after two or three doses of the mRNA Pfizer-BioNTech vaccine against SARS-CoV-2 and to investigate clinical features associated with non-response. **Material and methods:** 179 patients with LC of CTP class A in 93.3% and viral etiology in 70.1% of cases were longitudinally evaluated starting from the day before the first dose to 4 weeks after the booster dose. Their antibody responses were compared to those of healthcare workers without co-morbidities. In a subgroup of 40 patients, B- and T-cell responses were also compared to controls. **Results:** At d31, d90 and d180 after BNT162b2 vaccine, no detectable SARS-CoV-2 IgG response was observed in 5.9%, 3.9% and 7.2% of LC patients as compared to 0 controls (*p* < 0.03). A delay in B-cell and lack of prompt T-cell response compared to healthcare workers was also registered. A significant correlation between antibody titers and cellular response was observed. A MELD score > 8 was the only independent predictor of poor d31 response (*p* = 0.028). **Conclusions:** Our results suggest that cirrhotic patients have a slower and in <10% suboptimal immune response to SARS-CoV-2 vaccination. Rates of breakthrough infections were comparable between cirrhotics and controls. The booster dose was critical in inducing both humoral and cellular responses comparable to controls.

## 1. Introduction

Two mRNA-based technology vaccines, BNT162b2 and mRNA-1273, were developed by Pfizer-BioNTech and Moderna, respectively [1,2]; both were shown to be efficient in protecting against SARS-CoV-2 related disease [1,2]. In March 2021, the Italian Health Ministry authorized the vaccination of frail populations with two doses of mRNA vaccines, 3 or 4 weeks apart [3]. Given the higher risk of COVID-19-related mortality in patients with advanced liver disease compared to the general population [4], EASL and AASLD guidelines recommended prioritizing for vaccination patients with advanced liver diseases or patients who had received a liver transplant [5,6].

Cirrhotic patients are characterized by alterations in the immune system and consequent vaccine hypo-responsiveness. As shown after hepatitis A and B vaccinations, an inverse relationship between the severity of LC and vaccine response can be observed [7,8]. Currently available data on SARS-CoV-2 vaccine response in patients with LC-included “de facto” among frail subjects- are currently limited [9]. Indeed, in the clinical BNT162b2 trial from Pfizer-BioNTech leading to the mRNA vaccine registration over 43.448 randomized to vaccine or placebo; no more than three patients (<0.1%) with severe liver disease were included [1]. Similarly, only 196 (0.6%) subjects with no better defined liver disease out of 30,420 subjects randomized to mRNA-1273 or placebo received the Moderna vaccine in the registration trial [2].

A few studies have specifically explored SARS-CoV-2 vaccine response in cirrhotics. A small cohort study conducted in the US focused specifically on liver transplant patients, evaluating humoral response in 79 patients with LC vaccinated with both mRNA vaccines or with a single dose of the Johnson and Johnson DNA vaccine and showed differences in antibody response by vaccine type [10]. The study reported a poor response in about 24% of patients with chronic liver diseases with or without LC and in 61.3% of liver transplant patients, although LC did not result in an independent predictor of poor humoral response [10]. The study lacks follow-up data and a control group. Another prospective observational study, conducted in Hamburg, reported seroconversion after the second vaccine dose in 100% of 48 cirrhotic patients. In more detail, in this study among those vaccinated with PfizerBioNTech, only 38 with LC, mostly related to alcohol abuse, were included. The immunological response of these patients was evaluated after a short post vaccination follow-up of only 28 days [11] suggesting that data on long-term efficacy are still missing in this population. In this study, T-cell function tests were performed, but antigen-specific B-cell response was not investigated.

Recently, breakthrough infections, defined as SARS-CoV-2 infection after a full vaccination cycle, have been reported at increasing rates [12,13] and out of 459 subjects who received a solid organ transplant and both vaccine doses, 0.65% developed breakthrough infections [14], a rate higher than in general population.

This study aims at assessing, in a cohort of patients with LC compared to healthy controls of the same age, both humoral and cellular immune responses after two or three doses of the mRNA Pfizer-BioNTech vaccine against SARS-CoV-2, and at investigating the clinical features associated with non-response.

## 2. Patients and Methods

Consecutive 179 cirrhotic patients in Child Pugh Class (CTP) class A or B, vaccinated against SARS-CoV-2 with Pfizer-BioNTech between March and April 2021, and attending our outpatients clinic, were evaluated for humoral response at different time points after vaccination, including the booster dose received after a mean of 24 ± 2 weeks. In 40 of them who agreed to serial antibody testing at baseline, at d7 d31, d90 and d180 from the first dose, and 4 weeks after the third dose, 15 additional ml of blood were collected at these time points for cellular immunoresponse evaluation. Longitudinal humoral response results of cirrhotics were compared with those of healthcare workers (HW) from our institution without co-morbidities and a history of immunosuppressive therapy, vaccinated with the same vaccine schedule, and evaluated at the same time points. Cirrhotics and HW were matched. In order to avoid bias related to a different antibody response by age and gender, we subdivided our patients in quartiles and matched them by birth date ±1 year and gender in the ratio of 1:1 to a control group of HW. Overall, 117 previously SARS-CoV-2 uninfected controls of the same age were analyzed for antibody response and levels at the study time points.

The study was approved by the Ethics Committee of our Institution within the COVIDIAGNOSTIX project.

## 3. Antibody Analysis

Semiquantitative serological testing for IgG antibodies anti S1 domain (anti-S1) of SARS-CoV-spike protein was undertaken using the anti-SARS-CoV-2 QuantiVac enzyme-linked immunosorbent assay (EUROIMMUN, Lubeck, Germany) ELISA for IgG to the viral spike protein (S-protein) whose positive cutoff was of at least 3.2 BAU/mL. This assay was designed to evaluate vaccine response and calibrated against the WHO standard. The cut-off for positivity was 32.5 Binding Antibody Unit (BAU), low quantitation limit 3.2 BAU/mL at 1:101 dilution, range (3.2–384.0 BAU/mL). The result was 25.6 but <35.2 was considered borderline [15,16,17]. Specificity and sensitivity (>10 days after diagnosis) are 99.8% and 90.3%, respectively, when the manufacturers’ suggested cutoff of 35.2 U/mL (or BAU/mL) was used. A solution used for diluting samples above 348 U/mL was included in the measurement kits.

## 4. SARS-CoV-2–Reactive T Cells Analysis

Peripheral blood mononuclear cells (PBMCs) were isolated using standard procedures [18]. Briefly, 1 mL of EDTA-anticoagulated blood was diluted with 20 volumes of phosphate-buffered saline, pH 7.4 (PBS), containing 0.05 M ethylenediaminetetraacetic acid (EDTA; Invitrogen) and centrifuged for 5 min at 300× *g*. Afterwards, PBMC pellet was suspended in ammonium-chloride solution, incubated for 10 min at room temperature on a mixing platform in order to lyse contaminating red blood cells and centrifuged for 10 min at 300× *g*. Isolated PBMCs were washed with PBS-EDTA and subsequently resuspended in RPMI medium supplemented with 1 mM sodium pyruvate, 2 mM L-glutamine, 100 units/mL penicillin and 100 μg/mL streptomycin. PBMCs were stimulated in 100 μL in 96 wells plate at the concentration of 10^6^/well according to the manufacturer’s protocol for SARS-CoV-2 protein S T cell analysis (SARS-CoV-2 Prot_S T Cell Analysis Kit (PBMC), (Miltenyi Biotec, Bergisch Gladbach, Germany). The flow cytometry analyses were performed on FACS Canto2 (Becton Dickinson, Franklin Lakes, NJ, USA) (Appendix A). The FlowJo (Becton Dickinson, Franklin Lakes, NJ, USA) and GraphPad-Prism 8.4.3 software were employed for the visualization and statistical analyses of data. Gating strategy for identifying the different T-cell subsets in peripheral blood mononuclear cells (PBMCs) after in vitro antigen stimulation is reported in Appendix A.

## 5. SARS-CoV-2–Interacting B Cells Analysis

To detect SARS-CoV-2–interacting B cells, 50 μL of EDTA-anticoagulated peripheral blood were incubated with 500 μL of ammonium chloride solution. After red blood cell (RBC) lysis, cells were centrifuged for 10 min at 300× *g* and subsequently washed with PBS/3%FBS. Afterwards, as previously reported [18], cells were resuspended in 200 μL of PBS with 35 μg of protein extracts, containing the recombinant sfGFP-tagged SARS-CoV-2 S/RBD protein or sfGFP only as a control. After 20 min of cell protein incubation at room temperature, cells were washed with PBS and then stained with an anti-human CD45 eFluor 506-conjugated antibody (BDBioscience), an anti-human CD19 PE-conjugated antibody (BDBioscience), an anti-human CD38 PECy5-conjugated antibody (BDBioscience) and an anti-human CD3 SB436–conjugated antibody (BDBioscience, Franklin Lakes, NJ, USA). The DRAQ7 fluorescent DNA dye (1:1000 dilution; ThermoFisher, Waltham, MA, USA) was included to identify the fraction of total living cells Markers used in the flow cytometry assay are reported in Appendix A.

## 6. Statistical Analysis

Data are displayed as number and percentage for categorical variables. Continuous variables are expressed as mean and standard deviation (SD) or median and interquartile range (IQR) for normally distributed variables. Non-normally distributed variables were compared by the *t*-test and the Mann–Whitney U test when comparing two groups or by the Kruskal–Wallis test when comparing more than two groups, respectively.

Anti S1antibody dichotomous levels were compared by Chi-square test or Fisher’s exact test as the other categorical variables. To identify variables significantly contributing to the humoral and cellular response, a binary logistic regression model was constructed based on rational assumptions to predict a positive immune response. A set of independent variables (gender, age, esophageal varices, Child-Pugh classes, HCC diagnosis and etiology) was used to find the best fitting model to describe the relationship between independent variables and humoral immune response. A two-sided *p* value < 0.05 was considered to be statistically significant. Analyses were performed using Matlab statistical toolbox version 2008 (MathWorks, Natick, MA, USA) for 32 bit Windows.

## 7. Results

The demographics of 179 cirrhotic patients are reported in Table 1. In total, 61.0% were male. Median age was 68.0 (59.7–74.5). One hundred and sixty-four were in Child-Turcotte-Pugh (CTP) class A, 15 in class B (8.4%). No patient in CTP C class was included. In total, 40 (22.3%) patients had esophageal varices of different grade of severity, 13 (7.2%) had HCC, and three had colon, prostate and lung cancer. Two patients had received a liver transplant. The most frequent etiology was past HCV infection. Combined with HBV and delta related infections, viral infections account for the vast majority of LC (71.5%). Only 16 cases were related to autoimmune disorders and rare cases to alcohol abuse.

As 40 participants agreed to blood sample collection in addition to antibody testing, we performed a comparison between the entire cohort and the subgroup of the 40 subjects who agreed to cellular response evaluation. No significant differences were observed between the two groups (Table 1). Therefore, we assumed that the subgroup is representative of the whole cohort. No liver transplant patients were included in this subgroup.

At baseline, 28 patients (15.6%) had antiS1 antibody titers higher than the assay’s positivity threshold, despite the absence of symptoms of SARS-CoV-2 infection in the past; they were considered previously infected and analyzed separately. Their characteristics are reported in Appendix A. There were no differences in clinical characteristics but for HCC, that was significantly less represented among the small group of COVID-19 experienced subjects.

## 8. Humoral Response Analysis of Previously Unexperienced Cirrhotics over Time

Analyzing 151 SARS-CoV-2 unexperienced cirrhotics in comparison to 117 age- and gender-matched HW, vaccinated at our institution as shown in Table 2, at d31, after two doses of the BNT162b2 vaccine, SARS-CoV-2 IgG response was observed in all but nine cirrhotics (5.9%). Among HW, at the same time point none had IgG results below the threshold (*p* = 0.0053). At d90, at variance with the control group, where none had antibody titers below the threshold, six patients (3.9%) had results below the assay threshold (*p* = 0.03) and the difference increased at d180 (*p* = 0.0048). Interestingly, three of six patients negative at d90 and three of 11 negative at d180 developed SARS-CoV-2 infection after a few months.

Mean levels of antibody response in 151 SARS-CoV-2 inexperienced cirrhotics, longitudinally analyzed, are reported in Figure 1 in comparison to the mean antibody titers observed over time in the control group. In patients with antibody levels higher than the assay threshold, at d7 and d21, titers were not different from controls. At d31, they were higher among controls (978.8 ± 737.8 vs. 878.3 ± 869.4), (*p* = 0.050). Interestingly, at d180, cirrhotic patients’ titers were slightly higher than in healthy controls (264.8 BAU/mL ± 206.2 vs. 190.3 BAU/mL ± 121.4, (*p* = 0.061).

Four weeks after the booster, a significant increase in humoral response was observed in 100% of the patients. Antibody titers were similar to those of HW 1103.7 ± 842.74 vs. 941.8 ± 806.77 (*p* = 0.24) (Figure 1).

## 9. Breakthrough Infections

A number of breakthrough infections were observed among either patients or controls. Subjects with a breakthrough infection were excluded from the previously reported analysis of d180 antibody titers. Breakthrough infections were numerically higher among cirrhotic patients than in controls as 10 of 151 (6.6%) developed breakthrough infections as compared to 7 of 117 (5.9%) controls (*p* = 0.83). None of them required hospitalization. Among 10 cirrhotic patients with breakthrough infections, three were from the subgroup with antibody levels below the assay threshold at d31.

## 10. Humoral and Cellular Response in SARS-CoV-2 Experienced

As shown in Appendix A, not unexpectedly, antibody response was significantly higher in patients with LC who had experienced SARS-CoV-2 infection in comparison to that of unexperienced cirrhotics at each time point. The difference was statistically significant at d7 with mean titers of 190.1 ± 64.5 BAU/mL in inexperienced as compared to 855.7 ± 396.9 BAU/mL in SARS-CoV-2 experienced (*p* = 0.016), and at d90 with titers of 395.0 ± 49.18 in inexperienced as compared to 958.5 ± 276.7 BAU/mL in prior infected (*p* = 0.002). Cellular response in cirrhotic SARS-CoV-2 inexperienced versus experienced is reported in Appendix A.

## 11. Longitudinal Evaluation of Antigen-Specific Cell Response

A schematic overview of an experimental approach for the detection of B cells interacting with SARS-CoV-2 recombinant S protein by flow cytometry is reported in Figure 2. (A) HEK-293T cells were transiently transfected with pcDNA3.1 vector, encoding the receptor binding domain (RBD) of SARS-CoV-2 protein S (spike) fused with a sfGFP fluorescent marker. Afterwards, the recombinant SARS-CoV-2 protein was purified after 2 days from transfection and employed for the immunostaining of human PBMCs after red blood cell lysis. Initially, cells interacted with the recombinant SARS-CoV-2 proteins or sfGFP only as a control. Subsequently after washing, cells were labelled with the reported B-cell panel of fluorophore-conjugated antibodies. (B) Flow cytometry plots of B cells from a representative sample after interaction with the recombinant sfGFP-tagged SARS-CoV-2 S/RBD protein, only sfGFP is a control and follows the blocking of binding with a native unlabeled SARS-CoV-2 S/RBD protein. A plot of non-B cells from the same sample after interaction with the recombinant sfGFP-tagged SARS-CoV-2 S/RBD protein is also reported. B cells were determined in the CD19 + CD3- cell fraction. A Boolean gate was applied to identify non-B cells using the FlowJo (Becton Dickinson) software. (C) Overview of a gating strategy for identifying the different B-cell subsets in peripheral blood mononuclear cells (PBMCs). Fluorescence minus one (FMO) controls were used to set up all gates. Singlets were initially discriminated on SSC-H and SSC-A, followed by the exclusion of non-viable cells with Live/Dead far-red fluorescent DNA dye and the identification of CD45+ cell fraction. B cells were identified as CD3-CD19+ and plasma B cells were differentiated as CD38^high^ within the subset of B cells.

A subgroup of 40 cirrhotics, of whom 34 did not have any previous SARS-CoV-2 infection and six with previous exposure to SARS-CoV-2, adhered to additional blood sampling for T and B-cell studies.

Their immune responses are reported in Figure 3 and Figure 4 in comparison with HW evaluated at the same time points. In particular, a reduction of total and plasma antigen-specific B cells in seronegative cirrhotics with respect to the control group at day 21 (Total B cells: 0.028 vs. 0.055, Standard Error Mean (SEM) ± 0.011, *p* = 0.026; Plasma B cells: 0.083 vs. 0.725, SEM ± 0.06, *p* < 0.0001) and day 180 (Total B cells: 0.108 vs. 0.551, SEM ± 0.055, *p* < 0.0001; Plasma B cells: 0.14 vs. 0.428, SEM ± 0.06, *p* < 0.0001) was observed. In contrast, no differences were observed at day 7. These results suggest that patients with LC have a delay in the immune B cell response upon COVID-19 vaccination.

In order to determine the level of SARS-CoV-2–reactive T cells, a commercially available test for the identification of CD4+ and CD8+ T cell subsets after stimulation with a SARS-CoV-2 peptide pool was performed. In line with what was observed for antigen-specific B cells, CD4+ and CD8+ SARS-CoV-2–reactive T cells were boosted in the seronegative HW versus cirrhotics at day 21 (CD4+ T cells: 0.380 vs. 0.035, SEM ± 0.039, *p* < 0.0001; CD8+ T cells: 0.718 vs. 0.128, SEM ± 0.055, *p* < 0.0001) and day 180 (CD4+ T cells: 0.253 vs. 0.076, SEM ± 0.04, *p* < 0.0001; CD8+ T cells: 0.400 vs. 0.106, SEM ± 0.06, *p* < 0.0001). T cell response was similar in cirrhotics and controls at d7.

In keeping with the antibody response, B- and T-cell responses were enhanced in cirrhotics and controls 4 weeks after the booster (Figure 2 and Figure 3) However, after the booster, 12 cirrhotic patients were infected (7.9%) as compared to 25 HW (21.3%) (*p* = 0.002). All of them developed a paucisymptomatic infection. Only one was a subject that was poorly responsive through the complete vaccination cycle; of the remaining, another did not achieve antibody titers higher than the assay threshold, after the booster dose.

## 12. Comparison between Humoral and Cellular Immune Response

A direct correlation between antibody levels and changes in total and plasma antigen-specific B cells, as determined by the Pearson (r) correlation coefficients between S/RBD-interacting B cells and anti-SARS-CoV-2 IgG levels was observed (Appendix A). The correlation, statistically significant for each comparison, was more robust in HW than in cirrhotics.

Direct correlation between the levels of antibody titer and SARS-CoV-2–reactive T cells was also observed. The values for CD4+ and CD8+ T cells were r = 0.4688 (R^2^ = 0.2197) and r = 0.6366 (R^2^ = 0.4052) for HW and r = 0.3160 (R^2^ = 0.0998) and r = 0.3657 (R^2^ = 0.1338) for patients respectively (Figure 5).

## 13. Predictors of Poor Humoral Response

Among patients with results below the assay threshold at d31, a platelet count below 100.000 cells/mm^3^ (*p* = 0.44), evidence of esophageal varices (*p* = 0.01) and MELD score higher than 8 (*p* = 0.037) were negative predictors.

## 14. Discussion

This is the first study providing a complete longitudinal evaluation of humoral and cellular response to SARS-CoV-2 vaccine among patients with LC in comparison to a control group. In a small proportion of patients, less than 10%, mRNA BNT162b2 vaccine was unable to induce antibody response. Indeed, 31 days after the first vaccine dose, 6% of patients had undetectable antibodies and this rate increased to 7.2% 180 days after the first vaccine dose. None of age- and gender-matched controls had antibody response below the assay threshold at the corresponding time points (*p* = 0.0053 and 0.0058, respectively).

The highest antibody titers were observed at d31, although they were lower in cirrhotic patients than in controls. A progressive decline in antibody titers was observed thereafter in both groups until d180 whilst a significant increase was registered both in patients and controls after the booster.

In the subgroup of 34 cirrhotics, in the SARS-CoV-2 inexperienced—whose B- and T-cell immune response was longitudinally evaluated—B-cell response was shown to be impaired as compared to the controls until d180. The correlation between S/RBD-interacting B cells and anti-SARS-CoV-2 IgG levels was strong. Taken together, these data suggest that patients with LC have a delay in the B-cell response upon COVID-19 vaccination in subjects able to develop it and that B-cell response is boosted by the third vaccine dose. Only after the booster, both antibody levels and B-cell responses were similar to the control population. No certain correlates of protection have been so far identified in the context of COVID infection; however, of the 11 cirrhotics with antibody titers below the assay threshold at d180, one third developed SARS-CoV-2 infection before the booster.

As previously shown [19], immunophenotypic changes in T cell compartment are associated to SARS-CoV-2 infection, highlighting the relevance of T-cells in the innate and virus specific immune response. The evidence of the lack of prompt T cell response in cirrhotic patients as compared to HW is the other original finding of our study.

A direct correlation between the levels of antibody titer and SARS-CoV-2–reactive T cells reported in other studies [20] was also confirmed by our results. Notably, the level of significance of this correlation was higher in HW than in patients corroborating our finding suggesting not only a less durable, but also a slower response in vaccinated cirrhotics in comparison to the control group.

Documented vaccine hypo-responsiveness has been shown in liver cirrhosis not only after HBV vaccination, but also after pneumococcal and HAV vaccination [7,21,22]. We hypothesize that suboptimal liver synthesis of complementary and acute phase proteins might explain our observations. On the other hand, the cause of a delayed immune response in cirrhotic patients may be related to port-systemic shunts. Moreover, a few years ago, it has been reported that in HCV infected patients a loss of CD27+ memory cells contributes to suboptimal vaccine responses [23]. Studies so far conducted on SARS-CoV-2 vaccine-induced humoral responses have shown lower titers in liver transplanted patients and in patients with LC of CTP class C, mainly of alcoholic etiology [24]. Our findings in patients in CTP classes A or B suggest a possible B cell dysfunction even in patients with LC of less severe prognosis.

Two doses of BNT162b2 vaccine were shown highly effective against the delta variant of SARS-CoV-2, although slightly less than against the alpha variant [25] with unchanged protection from hospitalization [26]. However, as shown by a large-scale real world longitudinal study on HW in Israel, in the subgroup of immunosuppressed individuals, 6 months after BNT162b2 vaccination, the humoral response was reduced [27]. A booster dose was then proven able to reduce SARS-CoV-2 infection rate and severe COVID-19 illness in immunosuppressed individuals [28]. Our findings are in keeping with those results and demonstrate that all the hypo-responsive patients are able to show a solid humoral response, comparable to that of healthy individuals, only after the booster. Cellular response reflects the antibody kinetics and may be related to low cytokine response and T cell activation.

These results are well in agreement with other studies and with our own findings attained in HW [28,29] and strongly support the need for three-dose intensive vaccination in patients with LC.

Results observed in the subgroup of patients with previous SARS-CoV-2 infection were comparable to those reported in non-cirrhotic patients either in terms of antibody levels or in terms of strength of B and T cell responses [30].

Novel variants such as Omicron have the ability to evade B-cells but are associated with T-cell response, emphasizing the importance of analyzing B and T cell response of SARS-CoV-2 vaccinated subjects [31]. Of note after the arrival of the Omicron variant in our geographical area, during the third quarter of 2021, the number of the paucisymptomatic infections increased. Among cirrhotic patients who had completed the two-dose cycle, the number of infections was lower than that registered among HW maybe reflecting a lower exposure due to fear related to the increased mortality risk in the case of severe SARS-CoV-2 infection [32].

The availability of B and T cell responses, despite being evaluated in a small number of patients, is a strength of this study at variance with others recently published [33]. Another strength of the study is the long duration of follow-up and the availability of a matched control group, vaccinated with the same schedule. As advanced age may lead to an impaired immune response to the vaccine, regardless of liver disease stage, before reaching conclusions about immune response in cirrhotic patients, a control group of subjects of a similar age need to be considered.

Limitation is a lack of memory cell response evaluation. In addition, it should be considered that the measurement of T-cell responses in peripheral blood may not fully reflect what happens in the respiratory tract and lymph nodes.

In conclusion, taken together these results suggest that cirrhotic patients have a slower and suboptimal immune response to SARS-CoV-2 vaccination. B- and T-cell response mirrors antibody response. The booster dose appears to be critical in inducing humoral and B cell responses similar to those of healthy controls.

## Figures and Tables

**Figure 1 vaccines-10-01281-f001:**
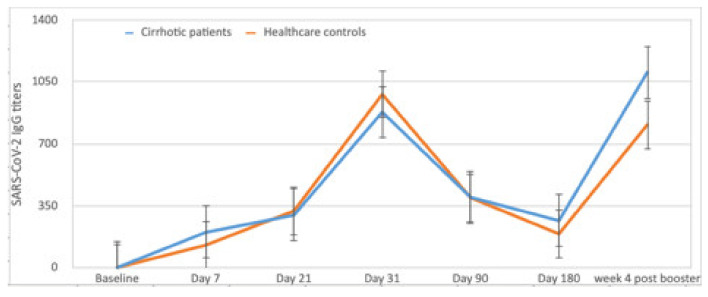
Longitudinal titers of SARS-CoV-2 IgG at the different time points in unexperienced cirrhotics and in HW. BAU/mL levels in patients with results higher than the positivity threshold are reported in y axis. Dose 1 was administered between 1 March and 15 March, dose 2 between 1 April and 15 April, dose 3 was administered 24 ± 2 weeks after the first dose.

**Figure 2 vaccines-10-01281-f002:**
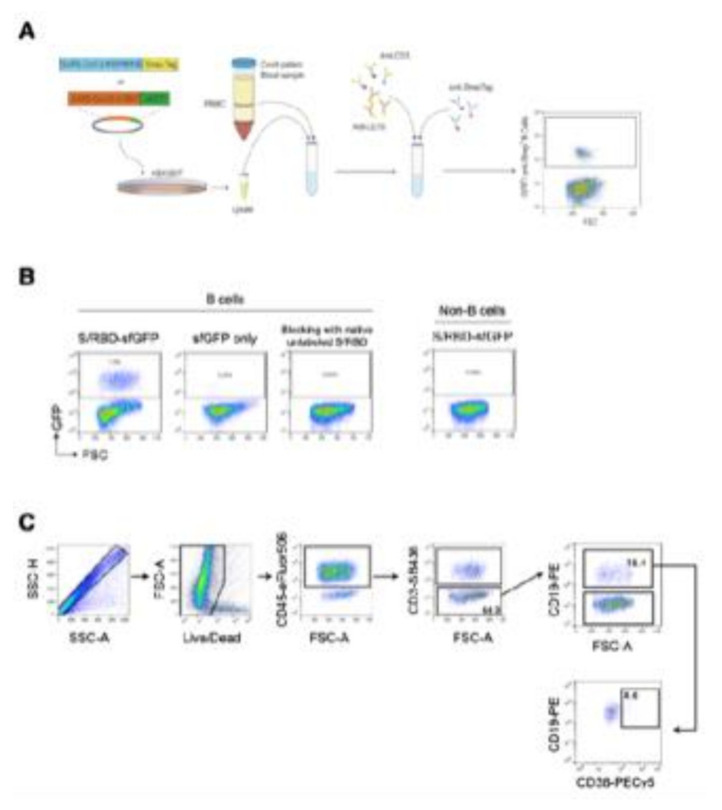
Schematic overview of experimental approach for the detection of B cells interacting with SARS-CoV-2 recombinant S protein by flow cytometry. (**A**) HEK-293T cells were transiently transfected with pcDNA3.1 vector, encoding the receptor binding domain (RBD) of SARS-CoV-2 protein S (spike) fused with sfGFP fluorescent marker. Afterwards, the recombinant SARS-CoV-2 protein was purified after 2 days from transfection and employed for the immunostaining of human PBMCs after red blood cell lysis. Initially, cells interacted with the recombinant SARS-CoV-2 proteins or sfGFP only as control. Subsequently after washing, cells were labelled with the reported B-cell panel of fluorophore-conjugated antibodies. (**B**) Flow cytometry plots of B cells from a representative sample after interaction with the recombinant sfGFP-tagged SARS-CoV-2 S/RBD protein, only sfGFP as control and following the blocking of binding with native unlabeled SARS-CoV-2 S/RBD protein. Plot of non-B cells from the same sample after interaction with the recombinant sfGFP-tagged SARS-CoV-2 S/RBD protein is also reported. B cells were determined in the CD19+CD3- cell fraction. Boolean gate was applied to identify non-B cells using the FlowJo (Becton Dickinson) software. (**C**) Overview of gating strategy for identifying the different B-cell subsets in peripheral blood mononuclear cells (PBMCs). Fluorescence minus one (FMO) controls were used to set up all gates. Singlets were initially discriminated on SSC-H and SSC-A, followed by the exclusion of non-viable cells with Live/Dead far-red fluorescent DNA dye and the identification of CD45+ cell fraction. B cells were identified as CD3-CD19+ and plasma B cells were differentiated as CD38^high^ within the subset of B cells.

**Figure 3 vaccines-10-01281-f003:**
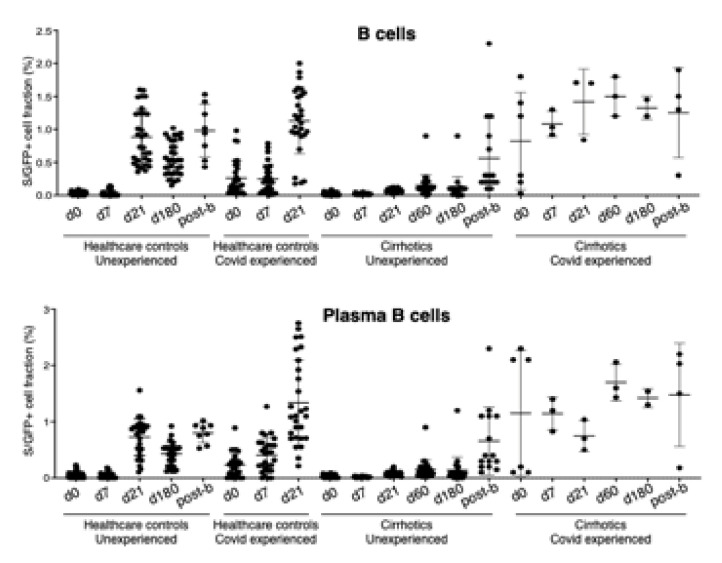
Flow cytometry assessment of total and plasma antigen-specific B cells in peripheral blood mononuclear cells (PBMCs) from HW and cirrhotics patients.

**Figure 4 vaccines-10-01281-f004:**
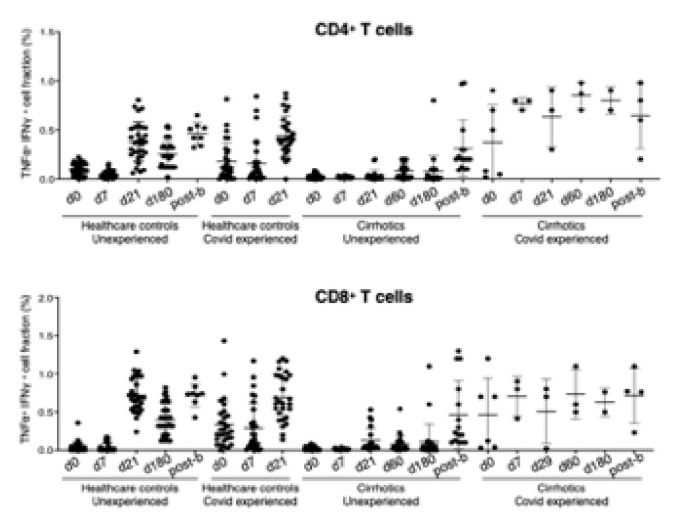
Flow cytometry assessment of CD4+ and CD8+ SARS-CoV-2–reactive T cells in peripheral blood mononuclear cells (PBMCs) from HW and cirrhotics patients.

**Figure 5 vaccines-10-01281-f005:**
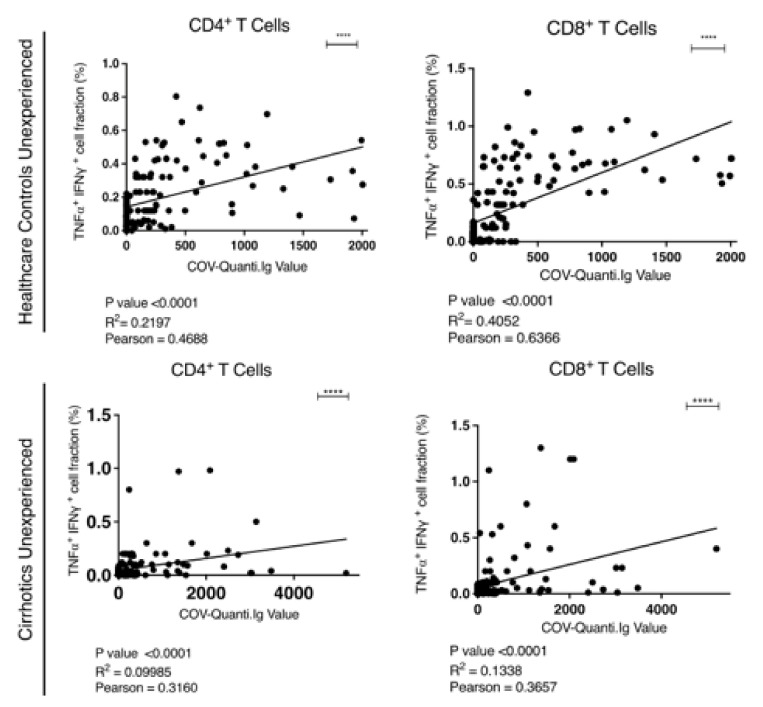
Correlation analysis between the anti-SARS-CoV-2 IgG levels and the abundance of CD4+ and CD8+ SARS-CoV-2–reactive T cells in peripheral blood mononuclear cells (PBMCs) from HW and cirrhotics patients. (**** indicates statistically significant difference).

**Table 1 vaccines-10-01281-t001:** Overall cohort of cirrhotic patients tested for humoral and subgroup tested for cellular immunoresponse against mRNA SARS-CoV-2 vaccine.

	Overall Cohort No. 179	Subgroup Tested for Cellular Response No. 40	*p* Value
Age, mean (SD), years	66.4 (11.1)	60.7 (10.4)	0.07
Median (IQR)	68.0 (59.7–74.5)	64.5 (51.5–69.0)	
Sex: Male	110 (61.4)	19 (47.5)	0.08
Female	69 (38.6)	21 (52.5)	
BMI, mean (SD)	26.9 (4.3)	26.6 (3.7)	BMI, mean (SD)
Etiology of liver disease			0.78
AIH/PBC/PSC	16 (9.0)	5 (12.5)	
HBV/HDV/HCV	128 (71.5)	30 (75.0)	
NAFLD	26 (14.5)	3 (7.5)	
Alcohol abuse	5 (2.8)	1 (2.5)	
Genetic hemocromatosis	4 (2.2)	1 (2.5)	
CTP class			0.74
A	164 (91.6)	36 (90.0)	
B	15 (8.4)	4 (10.0)	
MELD mean (SD)	8.6 (2.8)	8.4 (2.7)	MELD mean (SD)
PLT mean (SD)	153.2 (77.9)	131.4 (69.0)	PLT mean (SD)
HCC yes	13 (7.2)	5 (12.5)	0.56
HCC no	166 (92.8)	35 (87.5)	
Esophageal varices yes	40 (22.3)	16 (40.0)	0.18
Esophageal varices no	139 (77.6)	24 (60.0)	
Prior COVID infection yes	28 (15.6)	6 (15.0)	0.96
Prior COVID infection no	151 (84.4)	34 (85.0)	

**Table 2 vaccines-10-01281-t002:** Numbers and percentage of cirrhotic patients with antibody titers higher than the assay threshold.

	LC	HC Controls	*p* Value
	151	117	1.0
d7 No IgG < 35.2 BAU/mL, (%)	116 (76.8)	95 (81.1)	<0.0001
d21 No IgG < 35.2 BAU/mL, (%)	20 (13.2)	11 (9.4)	0.32
d31 No IgG < 35.2 BAU/mL, (%)	9 (5.9)	0	0.0053
d90 No IgG < 35.2 BAU/mL, (%)	6 (3.9)	0	0.03
d180 No IgG < 35.2 BAU/mL, (%)	11 (7.2)	0	0.0048

## Data Availability

Data are available at https://zenodo.org/record/5728042#.Ycy6BWnsJPw (accessed on 12 June 2022).

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
