# Peer review of "Effectiveness of Booster Dose of Anti SARS-CoV-2 BNT162b2 in Cirrhosis: Longitudinal Evaluation of Humoral and Cellular Response"

_vaccines, 2022, doi:10.3390/vaccines10081281_

Round 1

Reviewer 1 Report

Patients with chronic liver disease (CLD) and cirrhosis have worse outcomes from COVID-19 compared to those without liver disease. Therefore, liver societies have recommended vaccination against SARS-CoV-2 for all patients with chronic liver diseases to prevent severe and unfavourable consequencesSeveral SARS-CoV-2 vaccine platforms have been utilised among these populations. However, the immunogenicity, safety and efficacy data are variable and inconsistent due to a limited number of CLD, cirrhosis, and liver transplant patients included in the registration trials.

The manuscript by Giambra et a. assessed both humoral and cellular immune responses of patients with liver cirrhosis (LC) after two or three doses of mRNA Pfizer- BioNTech COVID19 vaccine. The longitudinal analysis of immune response was done in comparison with a cohort of matched controls.

The study is relevant because of the scarcity of data on vaccine-induced immune responses and efficacy data in cirrhotic patients who have received SARS-CoV2 vaccines. A suboptimal immune response is of concern, because suboptimal immunogenicity following vaccination against other infections in such patients is established and has been reported in the literature. The present study strongly supports the need of a three doses intensive vaccination in patients with LC and warrant further evaluation including need for additional booster doses, requirement for serologic testing to guide the need for booster vaccines, and effectiveness and safety of heterologous vaccines. These aspects and challenges should be better discussed in the manuscript.

The manuscript is scientifically sound and the experimental design appropriate. Strengths of the study are the long duration of follow up post vaccination, the assessment of B and T cell responses on top of humoral response and the availability of a well-matched control group, made of healthy HW vaccinated with the same vaccine and schedule. The major limitation, as acknowledged by the authors is related to the small cohort of patients assessed for cellular immune responses and the lack of evaluation of memory B and T cell response.

The method section is inaccurate and the paragraph describing the Antibody analysis has to be revised. More details have to be provided on the assay used, the cut-off and the units. As an example, 3.2 is not likely to be the cut-off but the conversion factor; BAU usually stands for Binding Antibody Unit and not – as in the manuscript – Binding Arbitrary Unit. The reference (#15) is not appropriate at least as a standalone.

The paragraph describing the method of SARS-COV2-interacting B cells will benefit from a description on how “B cells” and “plasma B cells” (or plasmablasts?) are differentiated using the different CD markers. The reference (#15) is not appropriate.

Figures/Tables as well as their legends in general are not accurate. Legend text is too short and not properly describing what is reported in the tables or legends.

Table 2 is difficult to interpret. The assay threshold is defined as 35.2 at variance with the method section. How was the threshold calculated?

Figure 1 is of poor quality and it just report mean levels. Error bars will have to be added as well as the y axis unit. The difference in the antibody response among LC patients and HW is not convincing also because of three different patterns at day 31, 90 and 180. The authors should better discuss this also in line with published data (reference #11) that show no significant difference in antibody responses between cirrhotic patients and healthy controls.

Moreover, the authors should discuss why they have measure only spike-binding antibodies and not neutralizing antibodies (NAs) that are an immune correlate of protection used to predict vaccine efficacy.

Supplementary tables/figures are cited in the text but were not part of the manuscript available to the reviewer.

It is not clear why the booster dose response is described in a separate paragraph given that data are reported in Figure 1, 2 and 3 and are part of the longitudinal analysis.

The conclusions are mostly consistent with the evidence and arguments presented. The most convincing data are the one generated by the analysis of B and T cellular immune responses in LC and HW controls that didn’t experience COVID infection. The analysis shows a delayed and impaired B and T cellular immune response in LC patients that is somehow restored only after the booster.

The references mostly cite recent publications and are relevant and appropriate apart from some in the Method section.  

Author Response

We thank reviewer #1 for the important comments and suggestions.

As requested the Methods section and in particular the Antibody analysis has been completely rewritten, and  B and T cell analysis description enriched (in the supplementary section).

Figure 1 legend is now more detailed and error bars have been added.

Table 2. has also been corrected.

To answer the Reviewer's question asking why neutralising antibody results were not shown, we can say that data on neutralising antibodies were available for all the Healthcare controls but only for a number of cirrhotic patients and we did not have enough resources to  complete the analysis.  Moreover,  as in a previous study from our group (Cellular and humoral immune responses and breakthrough infections after two doses of BNT162b2 vaccine in healthcare workers (HW) 180 days after the second vaccine dose. Front Public Health 2022:10:847384), in accordance with published data, a real correlate of protection was not clearly identifiable, in this study we preferred to explore B and T cellular responses.

Supplementary material is now correctly included in the submission.

In accordance with the Reviewer request the booster dose response is not anymore reported as a separate section but included in the humoral and cellular response sections, respectively.

Reviewer 2 Report

The goal of this study was to evaluate of humoral  and cellular (B- and T-cell responses) immune response in patients with liver cirrhosis versus healthy persons without comorbidity (control group) after two or three doses of mRNA vaccine (Pfizer-BioN-23 Tech) one to six months after vaccination. In this group of investigated liver cirrhosis patients (vs. control) a hypo-responsive (delayed) immune answer – that is a suboptimal immune response was observed. Also, a correlation between antibody titers and cellular response was registered. The use of booster doses of vaccine was important for sufficient/comparable initiation of humoral/cellular responses in cirrhosis patients (vs. control).

The phrase anti-SARS-CoV-2 is regularly in the use for specific antibodies (IgM, IgG, IgA), but not for vaccines (“…mRNA Pfizer-BioN-23 Tech anti-SARS-CoV-2 vaccine…”; line 24, 52, 86 …).

Is the “COVIDIAGNOSTIX project” maybe COVIDDIAGNOSTIX project? line 109

“…significantly significant…”? line 239

The paper does not provide an precise and sufficiently rational explanation for the obtained/presented results, why the booster dose gives more similar (comparable) immune responses (humoral and cellular) than first vaccination in the examined patients with cirrhosis and the control group. Perhaps it should be shown and discussed in more details. With the stated corrections, I believe that the paper could be published.

Author Response

We thank reviewer #1 for the advices.

The phrase anti-SARS-CoV-2 is regularly in the use for specific antibodies (IgM, IgG, IgA), but not for vaccines (“…mRNA Pfizer-BioN-23 Tech anti-SARS-CoV-2 vaccine…”; line 24, 52, 86 …) have been corrected.

The name of COVIDIAGNOSTIX project was correctly reported.

Significantly has been changed into statistically significant.

Two statements have been added to the Discussion in order to clarify the interpretation of the suboptimal immune response  to mRNA vaccine against SARS-CoV-2 in patients with cirrhosis. The sentences are reported in red.

Reviewer 3 Report

In the current manuscript, Giambra et al assessed the humoral and cellular immune responses in cirrhotic patients following two or three doses of the BNT162b2 COVID-19 vaccine. The authors show that, compared to healthy controls, an increased frequency of cirrhotic patients failed to develop antibody titers above the assay threshold at day 31, day 90, and day 180 post-vaccination. Further, the authors report that cirrhotic patients show a delay in SARS-Cov-2 specific B-cell and T-cell responses compared to healthy controls. They subsequently show that B-cell responses in cirrhotic patient are boosted by a third vaccine and that only after a third dose do antibody levels and B-cell responses in cirrhotic patients reach those observed in healthy controls.
The study described in the current manuscript was appropriately designed and the experimental data shown were largely convincing. However, there are some issues that need to be addressed and edits required to improve this manuscript:
Major comments:
1. It is essential that representative flow cytometry plots be included in this manuscript as supplementary data. The validity and significance of the authors’ findings would be strengthened by their showing of the gating strategy used to identify B cells, plasma cells, and T cells; the amount of background staining (if any) observed between the sfGFP control and sfGFP tagged SARS-CoV-2 S/RBD protein; and representative quadrant plots used to generate the numerical data shown in Figures 2 and 3.
2. The supplementary Tables and Figure mentioned in the text are missing.
3. For their antibody analyses, the authors state use of a “positive cut-off of at least 3.2 Binding Arbitrary Unit (BAU) ml as previously described (15)”. However, in Table 2 the authors report the number and % of patients with IgG <35.2 BAU/ml. This inconsistency in methodology and reporting needs correcting/explanation.
Minor comments:
1. Line 44: replace “frial” with “frail”
2. Line 94: the font size for the number 2 looks smaller than the surrounding text
3. Table 2: labeling in column one is inconsistent labeling. The authors use “No IgG <35.2 BAU/ml, (%)” for d7, d21, and d31 but switch to “IgG <35.2 BAU/ml, (%)” for d90 and d180.
4. Figure 1: it would be informative to the reader to indicate when on the timeline shown dose 1, dose 2, and dose 3 of vaccine was given. Alternatively, this information could be included in the figure legend.
5. Line 239: replace “significantly significant” with “statistically significant”
6. Line 283-286: font size is inconsistent with rest of text

Author Response

We thank reviewer # 2.

The supplementary material file has been revised and is now available (we apologise for previous missing files).

Flow cytometry plots are included in the manuscript as supplementary data. The  gating strategy used to identify B cells, plasma cells, and T cells is now clearly reported as well as the amount of background staining observed between the sfGFP control and sfGFP tagged SARS-CoV-2 S/RBD protein.  

The inconsistency in methodology about the threshold of positivity for the antibody assay used in this study have been corrected.

All the minor issues have been resolved.

Round 2

Reviewer 3 Report

I thank the authors for their timely reply. The authors have satisfactorily addressed almost all of my prior comments and I find the revised manuscript to be improved. I have just two comments on the revised manuscript:

1. Though the authors mention the inclusion of flow cytometry plots in the supplemental data those plots are missing from the submitted file. Likewise, mention of the supplemental flow cytometry data needs to be added to the main text (i.e., in the section titled "Longitudinal evaluation of antigen-specific cell response" or in the Methods section)

2. Figure 1 would be improved further with the addition of axis labels. While this is a minor point, I do think it would enhance the presentation of the data shown and be consistent with the elegant way the authors' other experimental data are presented in the manuscript (i.e., Figures 2-4).   

Author Response

We thank Reviewer#3

As requested the plots are now reported in Longitudinal evaluation of antigen- specific cell response.

We hope that Y Axis Title is now readable.